# Impact of Genetic Variations on Thromboembolic Risk in Saudis with Sickle Cell Disease

**DOI:** 10.3390/genes14101919

**Published:** 2023-10-09

**Authors:** Mohammad A. Alshabeeb, Deemah Alwadaani, Farjah H. Al Qahtani, Salah Abohelaika, Mohsen Alzahrani, Abdullah Al Zayed, Hussain H. Al Saeed, Hala Al Ajmi, Barrak Alsomaie, Mamoon Rashid, Ann K. Daly

**Affiliations:** 1King Abdullah International Medical Research Center (KAIMRC), Riyadh 11426, Saudi Arabia; 2King Saud Bin Abdulaziz University for Health Sciences (KSAU-HS), Ministry of National Guard Health Affairs (MNGHA), Riyadh 11426, Saudi Arabiaalzahranimo5@mngha.med.sa (M.A.);; 3Medical Genomics Research Department, King Abdullah International Medical Research Center (KAIMRC), Riyadh 11481, Saudi Arabia; 4Hematology/Oncology Center, King Saud University Medical City (KSUMC), Riyadh 11411, Saudi Arabia; falgahtani@ksu.edu.sa; 5Research Department, Qatif Central Hospital (QCH), Qatif 32654, Saudi Arabia; sabohelaika@moh.gov.sa; 6Pharmacy Department, Qatif Central Hospital (QCH), Qatif 32654, Saudi Arabia; 7King Fahad Hospital, Ministry of National Guard Health Affairs (MNGHA), Riyadh 11426, Saudi Arabia; 8Hematology Department, Qatif Central Hospital (QCH), Qatif 32654, Saudi Arabia; abalzayed@moh.gov.sa (A.A.Z.); alsaeedhha@hotmail.com (H.H.A.S.); 9Department of AI and Bioinformatics, King Abdullah International Medical Research Center (KAIMRC), Riyadh 11481, Saudi Arabia; 10Translational and Clinical Research Institute, Faculty of Medical Sciences, Newcastle University, Newcastle Upon Tyne NE1 7RU, UK

**Keywords:** sickle cell disease (SCD), thromboembolic events (TEEs), Saudis, GWAS, haplotypes, olfactory receptors

## Abstract

Background: Sickle cell disease (SCD) is a Mendelian disease characterized by multigenic phenotypes. Previous reports indicated a higher rate of thromboembolic events (TEEs) in SCD patients. A number of candidate polymorphisms in certain genes (e.g., FVL, PRT, and MTHFR) were previously reported as risk factors for TEEs in different clinical conditions. This study aimed to genotype these genes and other loci predicted to underlie TEEs in SCD patients. Methodology: A multi-center genome-wide association study (GWAS) involving Saudi SCD adult patients with a history of TEEs (n = 65) and control patients without TEE history (n = 285) was performed. Genotyping used the 10× Affymetrix Axiom array, which includes 683,030 markers. Fisher’s exact test was used to generate p-values of TEE associations with each single-nucleotide polymorphism (SNP). The haplotype analysis software tool version 1.05, designed by the University of Göttingen, Germany, was used to identify the common inherited haplotypes. Results: No association was identified between the targeted single-nucleotide polymorphism rs1801133 in MTHFR and TEEs in SCD (*p* = 0.79). The allele frequency of rs6025 in FVL and rs1799963 in PRT in our cohort was extremely low (<0.01); thus, both variants were excluded from the analysis as no meaningful comparison was possible. In contrast, the GWAS analysis showed novel genome-wide associations (*p* < 5 × 10^−8^) with seven signals; five of them were located on Chr 11 (rs35390334, rs331532, rs317777, rs147062602, and rs372091), one SNP on Chr 20 (rs139341092), and another on Chr 9 (rs76076035). The other 34 SNPs located on known genes were also detected at a signal threshold of *p* < 5 × 10^−6^. Seven of the identified variants are located in olfactory receptor family 51 genes (OR51B5, OR51V1, OR51A1P, and OR51E2), and five variants were related to family 52 genes (OR52A5, OR52K1, OR52K2, and OR52T1P). The previously reported association between rs5006884-A in OR51B5 and fetal hemoglobin (HbF) levels was confirmed in our study, which showed significantly lower levels of HbF (*p* = 0.002) and less allele frequency (*p* = 0.003) in the TEE cases than in the controls. The assessment of the haplotype inheritance pattern involved the top ten significant markers with no LD (rs353988334, rs317777, rs14788626882, rs49188823, rs139349992, rs76076035, rs73395847, rs1368823, rs8888834548, and rs1455957). A haplotype analysis revealed significant associations between two haplotypes (a risk, TT-AA-del-AA-ins-CT-TT-CC-CC-AA, and a reverse protective, CC-GG-ins-GG-del-TT-CC-TT-GG-GG) and TEEs in SCD (*p* = 0.024, OR = 6.16, CI = 1.34–28.24, and *p* = 0.019, OR = 0.33, CI = 0.13–0.85, respectively). Conclusions: Seven markers showed novel genome-wide associations; two of them were exonic variants (rs317777 in OLFM5P and rs147062602 in OR51B5), and less significant associations (*p* < 5 × 10^−6^) were identified for 34 other variants in known genes with TEEs in SCD. Moreover, two 10-SNP common haplotypes were determined with contradictory effects. Further replication of these findings is needed.

## 1. Introduction

Sickle cell disease (SCD) affects millions of people around the world but focuses more commonly on certain ethnicities in African, Caribbean, and Middle Eastern populations. A large proportion of Saudis, in particular people who live in the Eastern province (up to 24%), are affected by the disease or carry a single copy of the genetic trait [1,2]. It is an autosomal recessive disease caused by a point gene mutation: a transversion change in a single nucleotide base (rs334 A>T) in the codon region of the sixth amino acid of the hemoglobin β gene (HBB), located at chromosome (Chr) 11. This substitution results in an amino acid change from glutamine to valine, which ultimately encodes sickled hemoglobin, which polymerizes when deoxygenated and tends to produce abnormal (crescent-shaped) red blood cells (RBCs) [3]. The sickled reticulocytes (immature RBCs) in SCD have increased adhesive properties, which may trigger the vaso-occlusive process [4]. A clog of blood capillaries may lead to a disabling systemic syndrome, including chronic anemia, which may require frequent blood transfusions. Complications of SCD may also comprise difficult-to-treat leg ulcers, eye damage, stroke, vaso-occlusive (thrombotic) crises accompanied by acute pain, and organ infarction [5]. SCD patients may develop chronic organ malfunctions too, such as splenic sequestration crises, the formation of gallstones, and lung crises (the development of acute chest syndrome), that ultimately result in a poor quality of life, a poor prognosis, and a shorter life expectancy [6].

Several previous reports confirmed the increased risk of thromboembolic events (TEEs), both arterial and venous thromboembolism (VTE), in both SCD trait (HbAS) and SCD patients (HbSS) in comparison to control groups [7,8,9,10]. The reported level of risk in African-Americans with sickle cell trait is approximately twofold for VTE in general and fourfold for pulmonary embolism (PE) [11]. These results were confirmed recently in the UK, where sickle cell carriers were found to be at a higher risk of VTE, in particular PE (OR = 2.27), than healthy individuals [12]. In addition, in a study that involved 7000 SCD patients in the United States, the risk of developing PE was found to be fourfold the risk in patients without SCD [13]. Similarly, higher rates of infarctive cerebrovascular accidents and hemorrhagic strokes were noted in individuals with SCD than in normal subjects [14,15]. The prevalence of TEEs in SCD patients reported in African-Americans is about 11.3% to 11.5% [16,17]; this percentage is similar to the findings seen in the Saudi population (11.3%) [18], although a larger study showed a slightly lower prevalence (8.4%) [19]. Rates are extremely low in healthy individuals, approximately 1 per 1000 (0.1%) [20,21]. This might be attributed to several common risk factors, such as the placement of central venous catheters, obesity, pregnancy, and thrombophilias, or related to specific SCD factors, such as a history of splenectomy and the genetic makeup [22].

A number of reported candidate genes demonstrate a potential role in inducing thromboembolism in several clinical conditions. Factor V Leiden (FVL), prothrombin (PRT), and methylenetetrahydrofolate reductase (MTHFR) are the most common genes associated with a hypercoagulability status and are involved in various thrombophilic conditions. The risk of TEEs in women positive for the FVL (G1691A, rs6025) mutation increases by five times when they are exposed to oral contraceptives [23]. Thrombotic episodes and graft rejection were also noted in patients who underwent kidney transplantation and were heterozygous for the FVL variant [24]. Carriers of this variant are more susceptible to deep venous thrombosis (DVT) [25]. PRT (FII) (G20210A, rs1799963) was also identified as a risk factor for pulmonary embolism [26] and myocardial infarction [27]. Both variants in FVL and PRT were reported as the main contributing factors underlying recurrent pregnancy loss in Saudi females from different regions [28,29]. These two variants were also found to be more prominent in 250 TEE patients from Kashmir [30]. Furthermore, MTHFR variants (C677T (rs1801133) and A1298C (rs1801131)) were reported as risk markers in addition to FVL for developing DVT in two Iranian studies [31,32], although a previous study on a smaller number of patients failed to detect an association between DVT and MTHFR [33]. Different types of thrombotic conditions, such as portal vein thrombosis [34,35], postcardiac surgery thrombosis [36], and arterial thrombosis [37] were significantly associated with MTHFR, in particular the C677T variant. The polymorphisms in MTHFR are common in the healthy Saudi population; 23.9% are positive for C677T and 33.9% are positive for A1298C. In contrast, a lower number of individuals carry the variants G1691A in FVL (an average of 2%) and G20210A in PRT (2%) [38].

Previous comparative genetic studies between SCD patients and the normal population revealed variable allele frequencies of variants in the selected thrombotic genes. Two Indian studies that involved collectively 391 patients versus 447 controls [39,40] indicated a higher frequency of risk alleles in both the FVL and MTHFR (C677T) genes in patients compared to controls. However, the result failed to be replicated in 180 patients from the western region of India [41]. Two other studies conducted on Brazilian SCD patients showed a higher variant allele frequency in MTHFR (C677T) but not FVL in patients than in controls [42,43]. A recent study conducted on Tunisians showed more frequent polymorphisms in PRT (G20210A) and MTHFR (C677T) in patients with SCD compared with healthy subjects [44]. On the other hand, investigating the three polymorphisms in these genes (FVL, PRT, and MTHFR) in the Saudi population from the eastern province showed nonsignificant differences between 87 patients and 105 healthy individuals [45]. Due to the known role of these genes in the induction of thrombosis, we considered that we might see a higher rate of the selected mutations in SCD patients with a history of TEEs than in patients without a history of thrombosis. Thus, this study assessed the association of specific variants in FVL, PRT, and MTHFR with TEEs in Saudis with SCD and also performed a genome-wide association study (GWAS) to identify novel risk variants.

## 2. Methods

### 2.1. Study Design

This is a genetic association, case–control, multicenter study conducted to screen genes predicted to be associated with TEEs in SCD patients. The study was approved by the Institutional Review Boards of KAIMRC (Ref: IRBC/1414/19) and Qatif Central Hospital (QCH-SREC0216/2020). The recruited subjects were genotyped for selected variants in known thrombotic genes such as FVL (G1691A, rs6025), PRT (FII) (G20210A, rs1799963), and MTHFR (C677T, rs1801133). Moreover, a GWAS analysis comparing SCD patients with and without a TEE history was performed.

### 2.2. Sample Recruitment

Unrelated adult patients with SCD confirmed by a positive sickling test and homozygosity for the sickle mutation (rs334) (n = 350; 65 SCD patients with a history of TEEs plus 285 controls (SCD patients without a TEE history)) were recruited through their regular follow up with hematology units in three different settings: (i) King Fahad Hospital (KFH), Ministry of National Guard Health Affairs (MNGHA), Riyadh (n = 106 SCD patients; 27 cases with a TEE history plus 79 controls), (ii) King Khalid University Hospital (KKUH), King Saud University Medical City, Riyadh, (n = 82 SCD patients; 25 cases with a TEE history plus 57 controls), and (iii) Qatif central hospital (QCH), Eastern province, Qatif (162 SCD patients; 13 SCD with a TEE history plus 149 controls). Most of the patients who attended KKUH and KFH hospitals in the central region (Riyadh) were referred from southwestern or northern areas of Saudi Arabia. The average age of participants at the time of recruitment was 32.7 ± 10.2 years. Demographic and phenotypic data were collected for all participants. The focus in this study was on HbSS patients; thus, others with different types of SCD, such as HbSC, HbSbeta-thalassemia, HbSD, and HbSO, were identified via hemoglobin electrophoresis and were excluded. Exclusion criteria for TEE cases involved obese patients (BMI ≥ 30), smokers, females on contraceptive pills, pregnant women, history of postpartum, recurrent miscarriages, diabetes mellitus, Behçet’s disease, varicose veins, thrombophilia, chronic renal disease, ST elevation myocardial infarction (STEMI), cancer, immobilization, recent history of surgery, central venous catheter placement, admission at the intensive care units, trauma, fracture, recent long-distance travel, elevated levels of homocysteine, protein C, protein S, D-dimer, thrombin-antithrombin, and those with abnormal prothrombin time (PT, APTT). Only two cases fit these criteria and were excluded from our study cohort: a female with a postoperative DVT and another with a PE during pregnancy. Informed consent was obtained from the participants, and blood samples were collected from each patient for a DNA analysis.

### 2.3. TEE Diagnosis

Patients with TEEs were diagnosed through physician clinical judgment in conjunction with confirmatory imaging such as ultrasonography or computed tomography (CT). Stroke and transient ischemic attack (TIA) were confirmed via physical and neurological examination, e.g., electroencephalogram (EEG), laboratory (blood) tests, and imaging tests such as a doppler ultrasound, CT, or magnetic resonance imaging (MRI) scan.

### 2.4. Genomic Analysis

Total genomic DNA was extracted from whole blood using Puregene Blood Kits (Qiagen, Hilden, Germany, Catalog Number # 158389) according to the supplier’s instructions. An automated DNA extractor machine (KingFisher™ magnetic system, Thermo Fisher Scientific, Fresno, CA, USA) was used. A nanodrop (2000/2000c), Thermo Fisher Scientific, Fresno, CA, USA, was used to measure the absorbance of DNA at 260 nm. Working DNA stocks were aliquoted at a concentration of 50 ng/ul and stored at 4 °C. Cases and control samples were genotyped using the 10× Affymetrix Axiom array (Axiom 2.0 reagent kit designed by Applied Biosystems^TM^, Waltham, MA, USA, catalog number 901758), which includes 683,030 markers for the GWAS. The GWAS association analysis involved both common (minor allele frequency (MAF) ≥ 5%) and rare (MAF < 0.5%) variants. To ensure the validity and reliability of the used platform, a control sample with known variant calls was added to each test run. Axiom Analysis Suite software version 5.1 was used to cluster the genomic data; the average quality control call rate (CR) for the passing samples was 99.896%, and samples with >93% CR were retested. Genotype imputation was not performed here, as no reference panel is currently available for the Saudi population. The main five HBB haplotypes (Benin, Arab/Indian, Cameroon, CAR/Bantu, and SEN) can be ascertained through the genotyping of four SNPs (rs3834466, rs28440105, rs10128556, and rs968857) [46]. However, these SNPs are not included among the used GWAS panel. Thus, the association between HBB haplotypes and TEEs was not assessed.

### 2.5. Sample Size Calculation

Minitab version 16 was used to calculate the sample size needed as representative samples of the Saudi population to provide a statistical power of 80% at a 0.05 *p*-value cut-off significance. Our calculations based on MAF showed that a genotyping of 337 cases was needed for the FVL variant and 222 cases for the MTHFR variant to provide the suggested study power (Table 1). Therefore, we decided to recruit 350 cases in total.

### 2.6. Statistical Analysis

Plink software version 1.9 was used for the analysis of genomic data and generating linkage disequilibrium (LD) between SNPs. Samples with genotypes that were not in Hardy–Weinberg equilibrium (*p* < 0.05) were removed from the study. The *p*-values were calculated for categorical covariates such as genotype differences between different groups using Fisher’s exact test (Graphpad PRISM version 5.0). For GWAS, a *p*-value < 5 × 10^−8^ was set as a strict significance point to identify risk loci [47]. However, markers with *p* > 5 × 10^−6^ were also considered. Calculations of means and standard deviations were performed to assess age matching, and the differences were assessed using the 2-independent-sample *t*-test. Manhattan, quantile–quantile (Q-Q), and principal component analysis (PCA) plots were generated using the R statistical package (qqman) version R-4.2.2. The permutation test (T1) was used to assess sample heterogeneity based on pairwise identity-by-state (IBS) distance. The haplotype analysis software tool version 1.05, prepared by Eliades N-G. and Eliades D. G. [48], was used to determine the common significant haplotypes.

## 3. Results

As shown in Table 2, matching was confirmed for age (35.7 ± 9.8 years vs. 34.4 ± 10.3 years, *p* = 0.35) and sex (49.2% vs. 56.8% females, *p* = 0.27) between cases and controls, respectively. The majority of TEE cases had stroke (41.5%), PE (38.5%), and DVT (29.2%). The recruited patients with TEEs were mainly from the southwestern province (78.5%), with few cases from the eastern province (21.5%). A Q-Q plot of the observed versus expected *p*-values showed a late departure of the observed *p*-values from the null (Appendix A), indicating that the obtained results were not affected by genotyping quality, sample relatedness, or population stratification. Furthermore, the PCA plot indicated homogenous clusters of genetic variations that characterize cases and control cohorts (Figure 1). The genotype data of participants shown on the scatter plot revealed a matched background distribution. This was confirmed by the permutation test (T1) (*p* = 0.93), which ruled out the stratification effects between cases and controls. No significant difference was seen in the allele frequency of the MTHFR (rs1801133) polymorphism between the cases and control group (*p* = 0.74; Table 3). Moreover, the other two candidate genes, FVL (rs6025) and PRT (rs1799963), were not included in the analysis as the allele frequency of their variants was extremely low (<0.01). No meaningful comparison could be achieved in such a condition.

Five SNPs (rs2229637 in ITPR3, rs10998957 in LINC02651-RPL5P26, rs10746487 in H6PD-SPSB1, rs1985317, and rs6771316 in LINC00877) were previously tested in GWASs among French [49], African-Americans [50], and other mixed populations [51] that showed associations at *p* < 5 × 10^−6^ between the tested markers and venous thromboembolism (VTE). These results were replicated among our Saudi cohort but at lower significant levels (*p* = 0.016, 0.024, 0.028, 0.0496, and 0.044, respectively). Other variants in various genes were previously reported among the German population [52] (SLCO1B1 (rs4149056), PRIM1 (rs2277339), APOB (rs676210), TYK2 (rs12720356), TSEN15 (rs1046934), CYP4F2 (rs2108622), and MST1 (rs3197999)), and the Brazilians (ADAMTS (rs1364044)) [53] as risk factors for stroke in SCD patients with *p*-values < 1.0 × 10^−5^ in GWASs. These SNPs were tested in our stroke patients (n = 27), and no significant associations were detected except for rs1364044 in ADAMTS12 (*p* = 0.75, 0.74, 0.55, 1.0, 0.53, 0.55, 0.38, and 0.036, respectively).

Additional stroke markers (rs2084898 in TRIM29, rs469568 in ADAMTS2 [51], rs17696736 in SH2B3, rs4792143 in FLJ45455, rs16851055 in SPSB4, rs12646447 in PITX2, rs225132 in ERRFI1, rs161802 in PARK7 [54], rs12425791 in NINJ2 [55], rs2200733 in NR [56], rs556621 in SUPT3H [57], rs173686 in CSPG2 [58], and rs11853426 in ANXA2 [59,60]) were further tested here, but no associations with stroke cases in SCD were found except for the TRIM29 marker (*p* = 0.02, 0.65, 0.70, 0.81, 0.12, 0.07, 0.91, 0.88, 0.76, 0.08, 0.17, 0.32, and 0.06, respectively). The GWAS analysis of our data identified seven markers that surpassed genome-wide significance; five of them were located on Chr 11, (rs35390334 and rs331532 are in strong LD, rs317777, rs147062602, and rs372091), one SNP located on Chr 20 (rs139341092), and another on Chr 9 (rs76076035) (Figure 2). The SNPs rs317777 (an upstream variant in OLFM5P) and rs147062602 (a frame shift variant located in OR51B5) are exonic polymorphisms. In addition, we found 34 additional SNPs located on different genes with an association threshold of *p* < 5 × 10^−6^ (Table 4). Most of them were located at Chr 11, and some resided in potential sites such as rs73395847 (a splice acceptor variant in the C11orf40 gene) and rs80034548 (a noncoding transcript exon variant in the OR51A1P pseudogene). In addition, rs6554634 is a transcription factor binding site variant in the SLC6A19 gene at Chr 5, rs7933549 in OR51V1 is a missense variant, and rs73402629 in HBG1 and rs2071348 in HBBP are transcript enhancers located at the human β-globin locus. Two of the detected intronic variants (rs1455957 and rs11035718), which were in a perfect LD (r^2^ = 0.9, Appendix A), were both found to be in a strong LD (r^2^ = 0.77 and 0.85, respectively) with a missense variant (rs2472530) in the OR52A5 gene. These SNPs were all located at Chr 11. An SNP (rs5006884-A) in OR51B5 was less frequent in the cases than in the controls (MAF = 0.15 vs. 0.27, *p* = 0.003, OR = 0.46) and interestingly, the cases showed significantly lower levels of HbF compared to the controls (mean ± SD = 10.7 ± 8.0 vs. 14.1 ± 7.6, *p* = 0.002), Table 2.

The subgrouping analysis, which included stroke patients only, showed significant associations beyond the GWAS threshold of *p* < 5 × 10^−8^ with 84 variants; 26 of them were among the identified 41 loci detected for all TEE patients, though the associations seen in stroke cases were drastically stronger; see Table 4 and the Appendix A. A haplotype analysis was conducted, focusing on the top 10 SNPs with independent significance (no or weak LD) at a GWAS threshold of <1.0 × 10^−6^. The haplotype inheritance pattern of the selected markers (rs35390334, rs317777, rs147062602, rs4910823, rs139341092, rs76076035, rs73395847, rs1368823, rs80034548, and rs1455957) emphasized 131 haplotypes. The common haplotypes (observed in ≥5 individuals) were 14 (Table 5), and 2 of them indicated significant associations: a risk haplotype (haplo-91: TT-AA-del-AA-ins-CT-TT-CC-CC-AA) and its protective inverse (haplo-22: CC-GG-ins-GG-del-TT-CC-TT-GG-GG) with TEEs in SCD (*p* = 0.024, OR = 6.16, CI = 1.34–28.24, and *p* = 0.019, OR = 0.33, CI = 0.13–0.85, respectively).

## 4. Discussion

Major progress has been made in the genetics field since the advent of GWASs, which allows genetic testing across the whole genome with improved resolution to identify variations with the highest level of association [61]. A GWAS usually requires thousands of cases and controls to detect modest to strong associations with sufficient statistical power [62]. However, for very strong associations, small (approximately 50 to 100) numbers of cases may suffice [63,64]. Various GWASs conducted on SCD patients have succeeded in identifying several distinct loci predicted to be additional phenotypic modifiers. For example, rs3115229 at Chr 4 showed a significant association with acute, severe vaso-occlusive pain in children with SCD [65]. Furthermore, GWAS data from two African cohorts showed significant associations between variable HbF levels and variants in GLP2R and near BCL11A and HBS1L-MYB [66,67].

A head-to-head single SNP comparison of variants located in selected thrombotic genes (FVL, PRT, and MTHFR) between our cases and controls showed no association with TEEs in SCD patients. The genotyping results related to MTHFR (rs1801133) were not consistent with TEE associations reported previously in other medical conditions among Iranians [31,32], Japanese [34], Italians [35], Americans [36], and Georgians [37]. This may indicate that the MTHFR variant has no general role in TEE susceptibility. In contrast, the genotyping comparisons for FVL and PRT variants in our study cohort were considered unreliable due to the extremely low frequency of their alleles (<0.01). In such a condition, inductive reasoning is not possible, and therefore the analysis of both variants was removed from the Section 3. These variants in FVL and PRT are known thrombotic factors impacting TEEs but not in SCD across various ethnic groups, including the Saudi population, as supported by previous studies [23,24,25,26,27,28,29,30]. A recent meta-analysis that included 18 studies from mixed populations comparing 30,234 VTE cases and 172,122 controls detected FVL (rs6025) as the highest signal marker (1.4 × 10^−188^) [51]. The variant rs1799963 (in LD with rs191945075) in PRT was also reported with a very strong association (9.5 × 10^−32^). Thus, our data do not rule out the possibility of the involvement of both markers in TEE development. Multiple variants in the SLCO1B1, PRIM1, APOB, TYK2, TSEN15, CYP4F2, and MST1 genes were previously suggested as risk factors for stroke with *p*-values < 1.0 × 10^−5^ in a GWAS on German pediatrics [52]. Our study on a larger number of adult stroke patients (n = 27) revealed no associations between the suggested markers and stroke. In contrast, the association between rs1364044 in ADAMTS and stroke in SCD, reported in pediatric patients from Brazil [53], was detected among our adult Saudis with stroke and SCD (*p* = 3 × 10^−6^ vs. 0.036, respectively). This finding may imply a role for the variant in stroke induction, specifically in SCD. An association suggested by Arning et al. for another stroke marker (rs2084898 in TRIM29) [52] was also confirmed in our study. Furthermore, GWAS signals at the *p* < 5 × 10^−6^ threshold for five SNPs (rs2229637 in ITPR3, rs10998957 in LINC02651-RPL5P26, rs10746487 in H6PD-SPSB1, rs1985317 in an intergenic region, and rs6771316 in LINC00877) were reported previously with TEEs [49,50,51]. These signals were detected in our study as well, but at lower association levels. This replication provides evidence that the reported associations are not chance findings.

Genome-wide significant or close to significant associations with 41 markers were identified in the current study, with many of them (73.2%) located on Chr 11. SCD is a monogenic disorder involving Chr 11 [3], with various complications also predicted to be partially influenced by parental genetics. Thus, it is not surprising that the majority of genetic modifiers were seen in Chr 11. Four of the top detected signals that met the GWAS significant threshold cut-off are intronic variants with no previous phenotypic association reports. Introns are noncoding loci; however, some intronic variants may play a role in manipulating gene function via disrupting the RNA splicing process that takes place at exon–intron boundaries [68].

Our data also showed a possible association with rs317777 in a pseudogene (OLFM5P). Pseudogenes are DNA segments that have lost their coding ability; hence, they have no direct impact on phenotype occurrence but have the potential to influence the expression and activity of other coding genes possibly related to phenotypic diversity [69]. The associations seen with variants in olfactory receptor genes, family 51 (OR51B5, OR51V1, OR51A1P, and OR51E2) and family 52 (OR52A5, OR52K1, OR52K2, and OR52T1P), represented 29.3% of the suggested risk variants. The results highlighted frameshift and missense variants in two olfactory receptor genes (rs147062602 in OR51B5 and rs7933549 in OR51V1). These genes are homologous (paralog) and have closely similar structures and functions [70]. A frameshift variant is an insertion or deletion of a number of base pairs that generates a stop codon and usually causes a premature termination of a DNA sequence [71]. Previous reports have identified two variants in the olfactory receptor region associated with various phenotypes: rs7948471-A in OR51B5 associated with a higher degree of hemolysis in SCD (*p* = 3 × 10^−10^), rs5006884-A in OR51B5 associated with an increased HbF level (*p* = 3 × 10^−8^) [72], and rs7950726-A in OR51V1 associated with variable HbA2 levels in healthy adults (*p* = 1 × 10^−11^) [73]. In line with these findings, our results confirmed the association of rs5006884-A in OR51B5, where it was significantly less frequent in the cases with lower HbF levels than in the controls. This may suggest a linkage between HbF levels and TEE development. More findings in our study signify the role of olfactory receptors on TEEs. For instance, two of the detected markers were found to have a strong LD with a missense variant (rs2472530) in another olfactory receptor gene (OR52A5). Similarly, another suggested risk variant (rs12286769) is located in the upstream transcriptional region near OR52K2. A different variant (rs569117290) in this gene was previously reported with variable mean corpuscular hemoglobin concentrations [74]. The mutations in the olfactory receptor region (OR genes) may modulate the chromatin structure of the CTCF binding site within the β-globin gene and interfere with the gene-receptor interaction [75]. Moreover, selected variants within OR genes may regulate the expression of HBG [76]. A few olfactory receptor genes, such as OR2L13, OR4D6, and OR1N1, were recently suggested as risk factors for a number of TEE incidents [77]. A functional analysis indicated an upregulation of the transduction pathway through which the signaling of olfactory receptors modulates platelet activation underlying TEE [77,78].

A splice acceptor variant (rs73395847) in the C11orf40 gene was among the top-detected markers in our study that could possibly impact TEE development. Several mutations in C11orf40 were reported previously in patients with fibromyalgia syndrome, but the gene function is not clearly known [79]. This study also suggested that two variants (rs73402629 in HBG1 and rs2071348 in the human β-globin locus (HBBP1)) might be relevant to TEE risk (*p* = 1.41 × 10^−6^ and 3.25 × 10^−6^, respectively). The variant rs2071348 was previously reported as a predictor for disease severity in anemic patients with β^0^-thalassemia/hemoglobin E (*p* = 3 × 10^−15^, OR = 4.05) [80], whereas a strong association between HBG1 (rs998870472) and low hemoglobin levels was recently reported (*p* = 1 × 10^−273^) [81]. HBBP1 (a pseudogene) and HBG1 are both involved in the interaction of the locus control region with globin genes, which is a critical step for γ-globin regulation in adults [82]. Furthermore, rs6554634 is a predicted transcription factor binding site located approximately 12 kilobases (kb) upstream of SLC6A19, denoted in our study as a possible marker for TEE susceptibility. Certain cis-regulatory elements can be found hundreds of kb away from the actual transcriptional site [83]. The variant rs6554634 is not currently reported in ClinVar (https://www.ncbi.nlm.nih.gov/clinvar/, accessed on 1 January 2023), but was suggested as a biomarker affecting patients’ response to cetuximab in the treatment of colorectal cancer (*p* = 1.76 × 10^−6^) [84]. Two haplotypes of independent markers showed significant associations: a haplo-91 with a sixfold risk of TEEs in the carriers and a haplo-22 with a protective effect. The haplotype analysis involved only the top ten independent risk loci. In addition, two-thirds of the suggested variants (n = 26) showed genome-wide significance when the GWAS analysis was restricted to stroke cases only. This may imply a role for these variants in the induction of thromboembolism. The recruitment of further cases may confirm this. Four common SNPs were previously suggested to assess HBB haplotype frequencies [46], but these SNPs were missing in our GWAS markers, so it was not possible to assess the haplotypes frequencies in our study cohort. A recent study on the Saudi population with SCD identified the common HBB haplotypes, where the Arab/Indian haplotype was predominant in the patients from the eastern province, whereas the Benin haplotype was most common in the southwestern individuals [85]. Moreover, the study showed higher incidents of stroke in 318 Southwesterners with SCD than in the 159 examined patients from the eastern region. A previous study on Egyptians indicated a higher risk of stroke in homozygous Benin/Benin than other haplotypes [86]. Consistent with this finding, we noticed a higher rate of TEE cases in the southwestern participants (34.4%), who are known to inherit Benin haplotype more frequently, in comparison to the eastern patients (8.4%).

This study included the largest genome-wide scan of Saudi SCD patients, and by considering markers that just failed to meet the normal GWAS threshold, it identified some interesting novel variants and a haplotype as possible risk factors; however, further work is needed to replicate the findings in an independent cohort to confirm that the detected associations are true signals and not related to technical or methodological bias [87]. Some of the detected signals in GWAS are located in noncoding regions, but these may influence the binding between enhancer elements and transcription factors, which ultimately modulate genes’ expression [88]. Functional studies are further needed to interpret GWAS findings and provide possible mechanisms through which the detected variants impact TEE development [89].

## 5. Conclusions

This study showed no impact of the known thrombotic gene, MTHFR, on TEEs induced in Saudi patients with SCD. However, significant GWAS associations, at levels of *p* < 5 × 10^−6^, were identified between 41 variants (30 of them on Chr 11) and TEEs in SCD. Seven of the markers showed novel and stronger associations (*p* < 5 × 10^−8^), two of them were exonic variants (rs317777 in OLFM5P and rs147062602 in OR51B5), but these findings need further replication studies to be confirmed.

## Figures and Tables

**Figure 1 genes-14-01919-f001:**
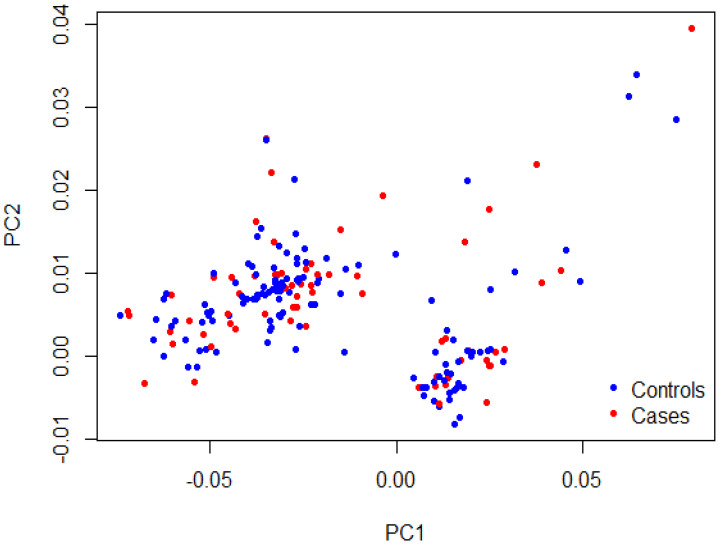
PCA plot showing sample stratification. The first two principal components (PC1 and PC2) of the cases in red and the controls in blue were plotted. The permutation test (T1) confirmed the samples’ heterogeneity (*p* = 0.93).

**Figure 2 genes-14-01919-f002:**
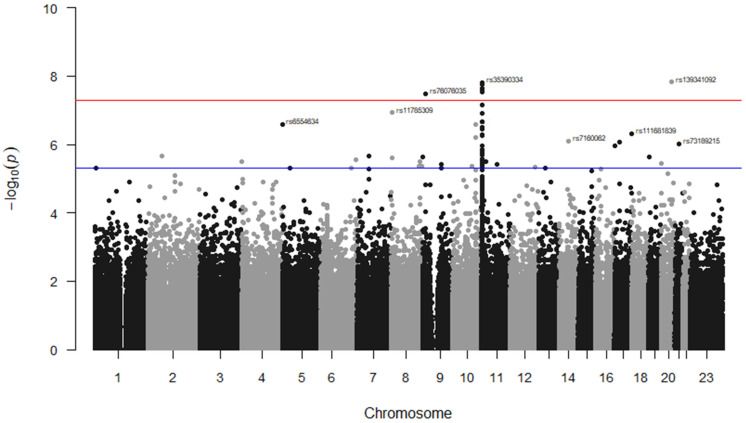
Manhattan plot for SNP associations with TEEs in the Saudi SCD cohort. The GWAS association *p*-values of the detected SNPs were plotted across the 23 chromosomes. Only the top marker on each chromosome was annotated. The SNPs which met the genome-wide significance threshold (*p* < 5 × 10^−8^) are shown just above the red line. The rs ID numbers of some SNPs with *p* < 5 × 10^−6^ (above the blue line) are demonstrated. The SNPs were plotted in two different colures (black and grey dots) to show a distinction between the chromosomes.

**Table 1 genes-14-01919-t001:** Sample size calculation based on MAF of FVL (rs6025) and MTHFR (rs1801133) at cut-off significance *p*-value of 0.05.

MAF Reported by Fawaz et al. [45]	FVL (rs6025)	MTHFR (rs1801133)
Controls	0.01	0.14
Cases	0.04	0.25
Sample size with 80% power	337	222

**Table 2 genes-14-01919-t002:** Demographic data of SCD patients who developed TEEs versus patients with no TEE history.

Variables	TEE Cases, n = 65 (%)	No TEE Controls; n = 285 (%)	*p*-Value
Males/Females	32/33	162/123	0.27
Age in years, Mean ± SD	35.7 ± 9.8	34.4 ± 10.3	0.35
Fetal hemoglobin (HbF), Mean ± SD	10.7 ± 8.0	14.1 ± 7.6	0.002
Southwestern province	51 (78.5)	133 (46.7)	
Eastern province	14 (21.5)	152 (53.3)	
DVT only	11 (16.9)		
PE only	18 (27.7)		
Stroke only	22 (33.8)		
DVT and PE	5 (7.7)		
DVT and stroke	3 (4.6)		
PE and stroke	2 (3.1)		
Other TEEs affected different organs (eye, ear, and liver)	4 (6.2)		

**Table 3 genes-14-01919-t003:** Association results of selected SNPs (reported previously as risk factors for thrombosis) with TEE cases in our SCD cohort.

	SCD Cases with TEEs (n = 65)	SCD Controls without TEEs (n = 285)	
Gene, Variant (Alleles)	Allele Frequencies (MAF%)		*p*-Value
*Known thrombotic gene*			
MTHFR, rs1801133 (A/G)	21/109 (0.16)	85/481 (0.15)	0.74
*Previously reported with TEEs*			
ITPR3, rs2229637 (A/G)	24/106 (0.18)	166/404 (0.29)	0.014
LINC02651-RPL5P26, rs10998957 (C/T)	18/112 (0.14)	131/435 (0.23)	0.020
H6PD-SPSB1, rs10746487 (T/C)	40/88 (0.31)	123/441 (0.22)	0.023
Intergenic, rs1985317 (T/C)	46/84 (0.35)	252/310 (0.45)	0.050
LINC00877, rs6771316 (A/G)	1/129 (0.01)	28/542 (0.05)	0.032

**Table 4 genes-14-01919-t004:** Markers with the highest association significance suggested to impact TEEs in SCD.

Variant	Chr	Variant Type (Gene)	Alleles	*p*-Value	
*Signals at a threshold of p* < *5 × 10^−8^ (n* = *7)*				*All TEEs*	*Stroke only*
rs139341092	20	Intergenic	CTCA/-	1.50 × 10^−8^	
rs35390334	11	Intronic	C > T	2.40 × 10^−8^	3.21 × 10^−9^
rs331532	11	Intronic	G > A	2.80 × 10^−8^	2.63 × 10^−9^
rs317777	11	Exonic (OLFM5P)	G > A	2.29 × 10^−8^	3.52 × 10^−11^
rs147062602	11	Exonic (OR51B5)	CCAGGTCTGTGGCAGCC/-	2.47 × 10^−8^	8.70 × 10^−11^
rs372091	11	Intronic (OR51B5)	G > A	2.95 × 10^−8^	4.43 × 10^−11^
rs76076035	9	Intronic	T > C	3.23 × 10^−8^	
*Signals located on known genes with an association threshold of p* < *5 × 10^−6^ (n* = *34)*					
rs4910823	11	Intronic (TRIM5, TRIM6-TRIM34)	A > G	7.14 × 10^−8^	3.71 × 10^−10^
rs11785309	8	Intronic (CSMD1)	T > C	1.18 × 10^−7^	2.67 × 10^−11^
rs73395847	11	Splice acceptor (C11orf40)	C > T	1.25 × 10^−7^	7.08 × 10^−10^
rs12412726	10	Intronic (LINC01435)	G > A	2.57 × 10^−7^	
rs6554634	5	Transcription factor binding site (SLC6A19)	G > A	2.64 × 10^−7^	
rs80034548	11	Exonic (OR51A1P)	G > C	3.08 × 10^−7^	1.95 × 10^−9^
rs1368823	11	Intronic (MMP26)	C > T	5.15 × 10^−7^	9.32 × 10^−9^
rs1455957	11	Linked with rs2472530 in OR52A5	A > G	5.37 × 10^−7^	4.70 × 10^−9^
rs73388885	11	3 prime UTR variant (OR51E2)	G > A	5.46 × 10^−7^	1.05 × 10^−9^
rs7933549	11	Missense (OR51V1)	G > A	1.08 × 10^−6^	3.17 × 10^−8^
rs73405065	11	Intronic (MMP26)	G > T	1.10 × 10^−6^	5.57 × 10^−9^
rs62071691	11	Intronic (CAMKK1)	A > G	1.10 × 10^−6^	
rs9667878	11	Upstream variant (OR51V1)	C > T	1.11 × 10^−6^	3.25 × 10^−8^
rs73402629	11	Upstream transcript enhancer (HBG1)	C > A	1.41 × 10^−6^	
rs11035718	11	Linked with rs2472530 in OR52A5	G > A	1.63 × 10^−6^	2.33 × 10^−8^
rs4525262	11	Intronic (TRIM22)	A > C	1.94 × 10^−6^	1.41 × 10^−10^
rs73405021	11	Intronic (MMP26)	A > G	2.06 × 10^−6^	2.24 × 10^−8^
rs317781	11	Intronic (UBQLN3)	A > C	2.06 × 10^−6^	9.84 × 10^−10^
rs35504021	9	Intronic (DOCK8)	G > A	2.36 × 10^−6^	
rs1566860	8	Intronic (CSMD1)	G > A	2.53 × 10^−6^	3.49 × 10^−8^
rs141682404	6	Intronic (PACRG)	C > T	2.74 × 10^−6^	
rs73402631	11	Upstream variant (HBG1)	C > G	2.98 × 10^−6^	
rs10742697	11	Upstream variant (UBQLN3)	C > A	3.16 × 10^−6^	1.63 × 10^−9^
rs2213170	11	Intronic (HBE1)	A > G	3.16 × 10^−6^	
rs2071348	11	Enhancer (HBBP1)	T > G	3.25 × 10^−6^	4.19 × 10^−9^
rs2599587	11	Intronic (TRIM21)	C > A	3.68 × 10^−6^	
rs73427713	11	Downstream variant (OR52K1)	A > G	3.80 × 10^−6^	4.43 × 10^−8^
rs7026337	9	Intronic (NTRK2)	A > G	3.93 × 10^−6^	
rs116926964	11	Intronic (ATG16L2)	G > T	3.93 × 10^−6^	
rs36030093	8	Intronic (LOC105375776)	C > T	4.21 × 10^−6^	
rs11033350	11	Intronic (OR51E2)	A > G	4.34 × 10^−6^	2.77 × 10^−8^
rs12286769	11	Upstream variant (OR52K2)	C > T	4.62 × 10^−6^	2.33 × 10^−8^
rs72981995	6	Intronic (LOC105378031)	A > C	4.85 × 10^−6^	
rs967664	11	Downstream variant (OR52T1P)	T > G	4.92 × 10^−6^	1.48 × 10^−8^

**Table 5 genes-14-01919-t005:** The common 10-SNP * haplotypes (observed in ≥5 individuals) seen in the tested SCD cohort of 65 cases and 285 controls.

Number of Observations	Haplotype	Haplotype Code	Inherited Cases (%)	Inherited Controls (%)	*p*-Value
11	CC-GG-ins-AG-del-TT-CC-TT-GG-GG	haplo-12	0 (0.0)	11 (16.92)	0.14
20	CC-GG-ins-GG-del-CT-CC-TT-GG-GG	haplo-17	2 (3.08)	18 (27.69)	0.39
8	CC-GG-ins-GG-del-TT-CC-TC-GG-GG	haplo-20	1 (1.54)	7 (10.77)	1.0
63	CC-GG-ins-GG-del-TT-CC-TT-GG-GG	haplo-22	5 (7.69)	58 (89.23)	**0.019**
5	CC-GG-ins-GG-ins-TT-CC-TT-GG-GG	haplo-25	1 (1.54)	4 (6.15)	1.0
5	TC-AA-del-AA-del-TT-TC-CC-CC-AA	haplo-40	0 (0.0)	5 (7.69)	0.59
5	TC-AG-W-AA-del-TT-TC-CC-CC-AA	haplo-50	0 (0.0)	5 (7.69)	0.59
5	TC-GG-ins-GG-del-TT-CC-TC-GG-GG	haplo-77	0 (0.0)	5 (7.69)	0.59
5	TC-GG-ins-GG-del-TT-CC-TT-GG-GG	haplo-78	0 (0.0)	5 (7.69)	0.59
15	TT-AA-del-AA-del-CT-TT-CC-CC-AA	haplo-84	5 (7.69)	10 (15.38)	0.17
23	TT-AA-del-AA-del-TT-TT-CC-CC-AA	haplo-89	8 (12.31)	15 (23.08)	0.051
7	TT-AA-del-AA-ins-CT-TT-CC-CC-AA	haplo-91	4 (6.15)	3 (4.62)	**0.024**
9	TT-AG-W-AA-del-TT-TT-CC-CC-AA	haplo-102	0 (0.0)	9 (13.85)	0.21
5	TT-AG-W-AG-del-TT-TT-CC-CC-AA	haplo-108	0 (0.0)	5 (7.69)	0.59

ins = insert, del = delete, W = wild-type copy (CAGCCCCAGGTCTGTGG), *p*-values in bold are significant. * The selected 10 SNPs are rs35390334, rs317777, rs147062602, rs4910823, rs139341092, rs76076035, rs73395847, rs1368823, rs80034548, and rs1455957.

## Data Availability

The detailed research data is unavailable due to privacy aspects.

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
