# Peer review of "Impact of Genetic Variations on Thromboembolic Risk in Saudis with Sickle Cell Disease"

_genes, 2023, doi:10.3390/genes14101919_

Round 1

Reviewer 1 Report

Methods

Sample recruitment:

1.     A positive sickling test is found in both sickle cell trait (HbAS, SCT) as well as in sickle cell disease (SCD). Individuals with SCT are not “patients with SCD.”

2.     The authors state that “The focus in this study was on HbSS patients; thus, others with different types of SCD, such as HbSC, HbSbeta-thalassemia, HbSD, and HbSO were excluded.” However, this would require a Hb electrophoresis, not a simple “sickling test” (e.g. SickleDex).

3.     Patients with TEE vs controls: The investigators excluded patients with many pro-thrombotic or somewhat high risk disorders (at least 19 by my count).  How many patients and controls were actually excluded? Unfortunately, such exclusions could actually reduce the sensitivity of the study, since many thromboses probably occur due to a multiplicity of risk factors acting simultaneously.

4.     What minor allele frequency was set for inclusion for analysis in the GWAS?

Results

5.     Stroke in a SCD patient is likely a vaso-occlusive event, not a thrombotic event, although thrombosis can also occur secondarily. 29 of 65 TEE patients had only stroke, while 10 additional patients had stroke + another thrombotic event. Data should at least be analyzed without the “stroke only” patients. (Table 3)

6.     Does Table 3 show allele frequencies or genotype frequencies (in which case homoxygotes and heterozygotes should be indicated)? The header says “Variant Gene” but perhaps that should be “Variant Allele”?

7.     What were the comparisons of MAR in this cohort compared to the studies associating genes with thrombosis in SCD in previous studies (presumably non-Saudi)? Could that account for lack of duplication?

8.     Consideration needs to be given to Hb haplotypes. These should be determinable via GWAS. Then they should be analyzed for association with measured outcomes as well as with presence of minor alleles.

Discussion

1.     Variants of the OR51B5 gene have been associated with HbF levels (Solovieff N  Blood 2010 v155:1815-1822).

2.     The authors should present LOD plots showing the linkage (or lack thereof) of variants on chromosome 11.

3.     The authors should point that that the MAFs of the known “thrombosis” genes FVL and PRT were extremely low and therefore probably shouldn’t have been included in the analysis. MTHFR is the only of the three that merited analysis.

Author Response

Please look at the attached file

Reviewer 2 Report

Alshabeeb et al. describe a genome-wide association study (GWAS) involving Saudi adult patients with SCD who had a history of TEE. The study highlights the potential importance of two novel genetic markers (rs35390334 and rs331532) in relation to TEE risk in SCD patients. Overall the manuscript is well written, the methods described adequately, and conclusions reasonable. I recommend that the manuscript be published with minor revisions to address a few typographical and grammar issues that exist.

A thorough proofreading of the manuscript will help catch typographical/grammar errors and rectify them. Examples of such errors are these sentences “Most of them are located at Chr 11, and some are reside in potential sites…”, and “The subgrouping analysis, which included stroke patients only showed significant associations beyond the GWAS threshold of P<5×10-8 with 50 varints…”

Author Response

Thank you very much for taking the time to review this manuscript. Please find attached the new version of the manuscript (the re-submitted file) with tracked revisions/corrections. 

Reviewer 3 Report

In their manuscript „Impact of genetic variations on thrombotic risk in Saudis with sickle cell disease“ Alshabeeb et al discuss the present their results of a genome-wide association study concerning concomittant sickle cell disease an thromboembolic events.

Major:

Table 3 describes an association of prothrombotic mutations (such as Prothrombin, Factor V Leiden and MTHFR). Still the table lacks absolute numbers in total.

The small case number (65 cases) may preclude a meaningful interpretation of the genotype data. Especially taking into account that other, non-genetical factors (medication, contraception, smoking, obesity, concomittant cancer disease) were not questioned before. Therefore it is not clear, if bias is shown in the small TEE-group.

In Discussion overlap of fibromyalgia and SCD is discussed which needs a lot of clarification, as fibromylgia does not at all comprise TEE or other vasoocclusive diseaese.

Minor:

Figure 2 needs reconfiguration as layout is not displayed properly.

Some improvement can be made.

Author Response

Thank you very much for taking the time to review this manuscript. Please find the detailed responses in the attached file and the corresponding revisions/corrections highlighted in the re-submitted manuscript.

Round 2

Reviewer 1 Report

Methods

Sample recruitment: The authors have satisfactorily resolved queries about sample recruitment and identification.

Results

1.     The authors revised their manuscript to exclude analysis of “stroke only” patients in their overall analysis.

2.     Table 3 labelling was altered to indicate that numbers indicated allele frequencies.

3.     RESPONSE 8:  The reviewer requested that outcomes be analyzed for association with measurable outcomes.  Perhaps the authors misunderstood what was being asked.  Hb haplotypes are generally indicated as Benin, Cameroon, CAR/Bantu, SEN, or Arab/Indian. These are determinable via GWAS. Then they should be analyzed for association with measured outcomes as well as with presence of minor alleles.

The haplotypes that are listed in Table 5 are not beta globin haplotypes. Instead, they are apparently haplotypes representing interactions of various genes with uncorrected P values. Thus, the data are interesting, although offer minimal additional information overall.

Discussion

1.     Variants of the OR51B5 gene have been associated with HbF levels (Solovieff N  Blood 2010 v155:1815-1822). While it is true that the authors have explained this, they have still failed to discuss how changes in HbF might influence the frequency of their measured outcomes. This should certainly be discussed in their Discussion. In fact, their failure to measure HbF levels is a distinct weakness fo this study, in light of their findings.

2.     LOD plots showing the linkage (or lack thereof) of variants on chromosome 11 are in the Supplemental material.

3.     The authors have now pointed out that that the MAFs of the known “thrombosis” genes FVL and PRT were extremely low and therefore were excluded in the analysis.

Author Response

Comment 1:

Hb haplotypes are generally indicated as Benin, Cameroon, CAR/Bantu, SEN, or Arab/Indian. These are determinable via GWAS. Then they should be analyzed for association with measured outcomes as well as with presence of minor alleles.

Response 1: Thank you for addressing this important aspect. We added the following in the methodology section:

The main five HBB haplotypes (Benin, Arab/Indian, Cameroon, CAR/Bantu, and SEN) can be ascertained through genotyping of four SNPs (rs3834466, rs28440105, rs10128556, and rs968857) (Shaikho et al., 2017), however, these SNPs are not included among the used GWAS panel. Thus, the association between HBB haplotypes and TEE was not assessed.

Also, we added in the discussion (1st paragraph, page 25):

Four common SNPs were suggested previously to assess HBB haplotype frequencies (Shaikho et al., 2017) [46], but these SNPs are missing in our GWAS markers, so it was not possible to assess the haplotypes frequencies in our study cohort. A recent study on the Saudi population with SCD has identified the common HBB haplotypes, where the Arab/Indian haplotype was predominant in the patients from the Eastern province, whereas the Benin haplotype was most common in the Southwestern individuals (Al-Ali et al., 2021). Also, the study showed higher incidents of stroke in 318 Southwesterners with SCD than in the 159 examined patients from the Eastern region. A previous study on Egyptians indicated a higher risk of stroke in homozygous Benin/Benin than other haplotypes (Abou-Elew et al., 2018). Consistent with this finding, we noticed a higher rate of TEE cases in the Southwestern participants (34.4%), who are known to inherit Benin haplotype more frequently, in comparison to Eastern patients (8.4%).

Comment 2:

Discussion

 Variants of the OR51B5 gene have been associated with HbF levels (Solovieff N  Blood 2010 v155:1815-1822). While it is true that the authors have explained this, they have still failed to discuss how changes in HbF might influence the frequency of their measured outcomes. This should certainly be discussed in their Discussion. In fact, their failure to measure HbF levels is a distinct weakness fo this study, in light of their findings.

Response 2: Thank you for pointing this out. We agree with this comment. Therefore, we compared HbF levels between cases and controls and identified the allele frequency of the previously reported SNP (rs5006884-A in OR51B5). The results showed significant proportional associations of the tested SNP and HbF levels with the occurrence of TEE.

To reflect this, we added a statement in the results section:

A SNP (rs5006884-A) in OR51B5 was less frequent in the cases than in the controls (MAF= 0.15 vs 0.27, P=0.003, OR=0.46) and interestingly the cases showed significant lower levels of HbF compared to the controls (Mean ± SD= 10.7 ±8.0 vs 14.1 ± 7.6, P=0.002), Table 2.

Also, in the discussion section we added the following:

 In line with these findings, our results confirmed the association of rs5006884-A in OR51B5, where it was significantly less frequent in the cases with lower HbF levels than in the controls. This may suggest a linkage between HbF levels and TEE development.

Abou-Elew HH, et al. (2018) β S globin gene haplotype and the stroke risk among Egyptian children with sickle cell disease. Hematology 23, 362-367.

Al-Ali AK, et al. (2021) Sickle cell disease in the Eastern Province of Saudi Arabia: Clinical and laboratory features. American journal of hematology 96, E117-E121.

Shaikho EM, et al. (2017) A phased SNP-based classification of sickle cell anemia HBB haplotypes. BMC genomics 18, 1-7.
